# Oncogenic Gαq activates RhoJ through PDZ-RhoGEF

**DOI:** 10.3390/ijms242115734

**Published:** 2023-10-29

**Authors:** Rodolfo Daniel Cervantes-Villagrana, Víctor Manuel Color-Aparicio, Alejandro Castillo-Kauil, Irving García-Jiménez, Yarely Mabell Beltrán-Navarro, Guadalupe Reyes-Cruz, José Vázquez-Prado

**Affiliations:** 1Department of Pharmacology, Cinvestav-IPN. Av. Instituto Politécnico Nacional, Col San Pedro Zacatenco, Mexico City 07360, Mexico; rcervantesv@cinvestav.mx (R.D.C.-V.);; 2Department of Cell Biology, Cinvestav-IPN. Av. Instituto Politécnico Nacional, Col San Pedro Zacatenco, Mexico City 07360, Mexico

**Keywords:** oncogenic Gαq, PDZ-RhoGEF, ARHGEF11, RhoJ, endothelial sprouting

## Abstract

Oncogenic Gα_q_ causes uveal melanoma via non-canonical signaling pathways. This constitutively active mutant GTPase is also found in cutaneous melanoma, lung adenocarcinoma, and seminoma, as well as in benign vascular tumors, such as congenital hemangiomas. We recently described that PDZ-RhoGEF (also known as ARHGEF11), a canonical Gα_12/13_ effector, is enabled by Gα_s_ Q227L to activate CdcIn addition, and we demonstrated that constitutively active Gα_q_ interacts with the PDZ-RhoGEF DH-PH catalytic module, but does not affect its binding to RhoA or Cdc. This suggests that it guides this RhoGEF to gain affinity for other GTPases. Since RhoJ, a small GTPase of the Cdc42 subfamily, has been involved in tumor-induced angiogenesis and the metastatic dissemination of cancer cells, we hypothesized that it might be a target of oncogenic Gα_q_ signaling via PDZ-RhoGEF. Consistent with this possibility, we found that Gα_q_ Q209L drives full-length PDZ-RhoGEF and a DH-PH construct to interact with nucleotide-free RhoJ-G33A, a mutant with affinity for active RhoJ-GEFs. Gα_q_ Q209L binding to PDZ-RhoGEF was mapped to the PH domain, which, as an isolated construct, attenuated the interaction of this mutant GTPase with PDZ-RhoGEF’s catalytic module (DH-PH domains). Expression of these catalytic domains caused contraction of endothelial cells and generated fine cell sprouts that were inhibited by co-expression of dominant negative RhoJ. Using relational data mining of uveal melanoma patient TCGA datasets, we got an insight into the signaling landscape that accompanies the Gα_q_/PDZ-RhoGEF/RhoJ axis. We identified three transcriptional signatures statistically linked with shorter patient survival, including GPCRs and signaling effectors that are recognized as vulnerabilities in cancer cell synthetic lethality datasets. In conclusion, we demonstrated that an oncogenic Gα_q_ mutant enables the PDZ-RhoGEF DH-PH module to recognize RhoJ, suggesting an allosteric mechanism by which this constitutively active GTPase stimulates RhoJ via PDZ-RhoGEF. These findings highlight PDZ-RhoGEF and RhoJ as potential targets in tumors driven by mutant Gαq.

## 1. Introduction

Oncogenic mutations in *GNAQ* generate constitutively active versions of Gα_q_, characterized by amino acid substitutions that generate GTPase-deficient variants (Gα_q_ Q209L/P/R/H/K/Y). These oncogenic mutants fail to hydrolyze GTP, maintaining an active conformation. In this work, we focused on Gα_q_ Q209L, which causes uveal melanoma and is less frequently found in cutaneous melanoma and other cancer types [1]. Other somatic mutations in *GNAQ*, particularly the one coding for Gα_q_ R183Q, cause endothelial malformations and tumors, as observed in Sturge–Weber syndrome [2,3]. Metastatic tumors caused by mutant *GNAQ* are highly lethal and, currently, there are no clinically successful therapies against them. Therefore, the characterization of the effectors of constitutively active Gα_q_ is necessary to identify potential pharmacologic targets and the development of effective precision treatments. The phosphoinositide/calcium/PKC pathway is activated by Gα_q_ through phospholipase Cβ, its prototypical direct effector, explaining many physiological effects of Gq-coupled receptors [4]. However, inhibition of key elements of this pathway is not enough to prevent the oncogenic actions of constitutively active GNAQ, indicating that unidentified effectors contribute to the oncogenic process and represent potential molecular vulnerabilities [5,6,7]. Synthetic lethality approaches revealed non-canonical pathways as the main drivers of oncogenic *GNAQ* signaling [8]. Recent phosphoproteomic and genome-wide synthetic lethality strategies and chemogenetic drug screening have revealed novel targetable signaling vulnerabilities in GNAQ-driven uveal melanoma [6,7,9]. To expand the possibilities to understand the molecular intricacies of unconventional pathways activated by oncogenic GNAQ, we focused our current studies on the hypothetical role of PDZ-RhoGEF/RhoJ as a direct signaling axis activated by Gαq Q209L, an oncogenic mutant, and used a rational data mining strategy, focusing on their relational signaling partners in cancer patients, to highlight those that, according to synthetic lethality datasets of cancer cells, represent vulnerabilities. 

PDZ-RhoGEF, also known as ARHGEF11, belongs to the family of RGS-RhoGEFs, which are characterized as the main effectors of Gα_13_-coupled receptors signaling to RhoA [10,11]. Downstream effectors of this pathway cause actin cytoskeleton reorganization into stress fibers and contractile actomyosin complexes [12]. We recently described that Gα_s_ Q227L enables PDZ-RhoGEF to activate Cdc42 via direct interaction with its catalytic domains (DH-PH) [13]. In addition, we found that constitutively active Gα_q_ also interacts with the DH-PH domains of PDZ-RhoGEF [13]. However, the functional consequences of this interaction remain to be deciphered. We postulate that GTPase-deficient Gα_q_ enables PDZ-RhoGEF to activate RhoJ, a member of the Cdc42 subfamily controlling endosomal trafficking of α5β1 integrins [14,15,16] and focal adhesion dynamics [17], as well as cancer progression via tumor-induced angiogenesis, metastatic dissemination of cancer cells, and drug resistance [14,15,16,18]. In this work, we addressed the regulation of PDZ-RhoGEF as a potential effector of constitutively active Gαq driven to gain affinity for RhoJ.

## 2. Results

### 2.1. PDZ-RhoGEF Directly Activates RhoJ 

Rho guanine nucleotide exchange factors (RhoGEFs) of the Dbl family are characterized by their complex structures, including multiple domains flanking a catalytic module composed of a Dbl-homology (DH) domain, followed by a Pleckstrin-homology (PH) domain [19,20]. To understand the mechanistic basis of RhoJ activation by these GEFs, we screened a group of EGFP-tagged catalytic domains (DH-PH) of different RhoGEFs. An isoprenylation signal was added to these constructs to be expressed as membrane-anchored constitutively active (CA) RhoGEFs (Figure 1A) [13,21,22]. We used GST-CRIB pulldown assays to identify those RhoGEFs able to activate RhoJ (Figure 1B) [21]. Since the regulation of some Rho GTPases might occur downstream of other Rho GTPases [23,24], we directly isolated active RhoJ-GEFs through pulldown with nucleotide-free GST-RhoJ-G33A. This way, we could identify GEFs in an active conformation with direct affinity for this small GTPase, a member of the Cdc42 subfamily [21]. We found that PDZ-RhoGEF (ARHGEF11) (Figure 1B–C), among other RhoGEFs including ITSN1 [21], β-Pix (ARHGEF7) (Figure 1B,D), and FARP2 (Figure 1B,E), were able to activate RhoJ and interacted with GST-RhoJ-G33A in transfected HEK-293T cells. We focused on the characterization of PDZ-RhoGEF as an activator of RhoJ, based on its hypothetical regulation by Gα_q_ Q209L, an oncogenic driver [1], which we recently identified as a previously unrecognized binding partner of the PDZ-RhoGEF’s catalytic module (PRG-DH-PH), but whose functional impact remained undeciphered [13]. The dominant negative RhoJ mutant (RhoJ-T35N), is characterized by an inactive conformation unable to be subjected to nucleotide exchange (Figure 2A). We hypothesized that, if the PDZ-RhoGEF’s catalytic domains were able to recognize and maintain a stable interaction with this inactive mutant, it would prevent the activation of endogenous GTPases by the constitutively active GEF. The interaction of PRG-DH-PH with the dominant negative RhoJ mutant was evaluated with a pulldown assay, using lysates of transfected HEK-293T cells. Indeed, the PRG-DH-PH module specifically interacted with RhoJ-T35N (Figure 2B), suggesting a potential inhibitory effect on PRG-DH-PH-mediated processes. As an initial readout of PRG-DH-PH cellular activity, we evaluated its effect on endothelial cell morphology. Furthermore, we addressed its potential link to RhoJ by testing whether the dominant negative RhoJ-T35N mutant was able to inhibit the morphological effects caused by constitutively active PDZ-RhoGEF (Figure 2A). Consistent with our previous findings [13], the membrane-anchored PRG-DH-PH construct caused contraction of endothelial cells and also generated fine sprouts. These morphological effects were prevented by the dominant negative RhoJ-T35N mutant (Figure 2C). RhoJ-T35N reduced the percentage of cells expressing PRG-DH-PH showing sprouting (Figure 2D) and contracted shapes (Figure 2E).

### 2.2. Gαq Promotes RhoJ Activation via Direct Interaction with PDZ-RhoGEF 

Previously, we showed that Gαs-Q227L interacts with PDZ-RhoGEF’s catalytic domains, leading to the activation of Cdc42 [13]. Furthermore, we demonstrated that Gαq-Q209L also exhibited an equivalent interaction, but did not activate Cdc42 [13]. Therefore, we hypothesized that Gαq-Q209L, unlike other active Gα subunits, could drive PDZ-RhoGEF to bind and activate RhoJ, since the latter is closely related to Cdc42 (Figure 3A). Consistent with this hypothesis, the interaction of PRG-DH-PH with GST-tagged RhoJ-G33A was promoted by Gαq-Q209L and not by other active Gα subunits (Figure 3B), suggesting that this pathway would lead to RhoJ activation. Moreover, the isolated complex of PRG-DH-PH with RhoJ-G33A also included Gαq-Q209L, suggesting that the interaction of this constitutively active GTPase with the PRG-DH-PH module drives this catalytic construct to recognize RhoJ (Figure 3B, HA-Gα blot in PD). Although constitutively active Gαs was also found in the RhoJ-G33A pulldown, it did not increase the interaction of PRG-DH-PH with RhoJ-G33A. This suggests that some unidentified endogenous guanine nucleotide exchange factor might mediate this interaction, a possibility that deserves future investigation. We then evaluated whether full-length PDZ-RhoGEF would act as an effector of Gα_q_ Q209L enabling the full-length RhoGEF to interact with RhoJ (Figure 3C). Using pulldown assays of cell lysates, we demonstrated that full-length PDZ-RhoGEF interacted with RhoJ-G33A (Figure 3D), and this interaction was more effective in the presence of the constitutively active Gα_q_ Q209L mutant (Figure 3E).

To gain insight into the structural basis of the interaction between Gα_q_ Q209L and PRG-DH-PH, we analyzed whether this constitutively active mutant Gα_q_ interacted with PDZ-RhoGEF constructs, including either the DH or the PH domains, or the linker joining them, to determine the minimal region of interaction (Figure 4A, left panel). Using GST-fused constructs of PDZ-RhoGEF’s DH, PH domains, and the linker region, we found that Gα_q_ Q209L was mainly interacting with the PH domain of PDZ-RhoGEF (Figure 4B). Based on these results, we hypothesized that the PH domain could compete the interaction of oncogenic Gα_q_ with PRG-DH-PH (Figure 4A, right panel). We used an EGFP-tagged PDZ-RhoGEF-PH (PRG-PH) construct to compete with the interaction between the HA-Gαq-Q209L and GST-PRG-DH-PH tandem. As predicted, the PRG-PH construct reduced the interaction between Gαq-Q209L and PDZ-RhoGEF’s catalytic module (Figure 4C). In contrast, a Gq inhibitor (YM254890), known to block Gq-coupled receptor signaling, but not the GTPase-deficient Gαq-Q209L mutant [25], was unable to prevent the same interaction (Figure 4A, right panel, and Figure 4D, iGq, 1 µM). We then evaluated whether the PH domain was able to disassemble the ternary Gα_q_-Q209L/PRG-DH-PH/RhoJ complex (Figure 4E). To address this question, we used a GST-RhoJ-G33A pulldown assay and looked for the Gα_q_-Q209L/PRG-DH-PH complex in control cells, and in cells co-transfected with the PRG-PH domain or treated with the Gq inhibitor (iGq, YM254890). Consistent with a competitive effect, the PH domain inhibited the interaction of Gα_q_-Q209L with the fraction of PRG-DH-PH bound to RhoJ-G33A. In contrast, the pharmacological inhibitor of Gq was unable to alter the complex (Figure 4E).

We then assessed whether immobilized RhoJ-G33A was able to capture constitutively active DH-PH constructs from the three members of the family of RGS-RhoGEFs (EGFP-DH-PH-CAAX from p115-RhoGEF, LARG, and PDZ-RhoGEF) (Figure 5A). Using pulldown assays with co-transfected GST-RhoJ-G33A (Figure 5B), we found that all three RGS-RhoGEF DH-PH modules interacted with RhoJ-G33A (Figure 5B). On the other hand, we analyzed whether oncogenic Gα_q_-Q209L was able to interact with these DH-PH constructs and found that Gαq-Q209L interacted with the DH-PH catalytic modules of LARG, PRG, and p115-RhoGEF (Figure 5C). However, experiments conducted to address whether these catalytic modules managed to activate RhoJ revealed that only PRG-DH-PH efficiently activated this GTPase, as assessed using pulldown with GST-PAK-CRIB linked to glutathione-sepharose beads (Figure 5D).

### 2.3. The Gαq/PDZ-RhoGEF/RhoJ Axis and Its Signaling Partners Correlate with Shorter Patient Survival of Uveal Melanoma Patients 

To identify possible regulators of the Gαq/PDZ-RhoGEF/RhoJ pathway we conducted an in silico analysis through data mining of the TCGA uveal melanoma study, which includes transcriptomic and clinical data from 80 patients [26]. We found 101 transcripts coding for signaling proteins that correlated with the expressions of at least two of the three genes of the Gαq-Q209L/PDZ-RhoGEF/RhoJ signaling axis, highly expressed in patients (Figure 6A). Thirty-three of them were correlated with shorter patient survival (indicated in bold). Fourteen of them, coding for GPCRs; catalytic signaling effectors, including kinases, phosphatases, small GTPases, and guanine nucleotide exchange factors; and non-catalytic signaling proteins (Figure 6B,D), were analyzed as transcriptional signatures. Expressions of *GNAQ*, *ARHGEF11, RHOJ* and the fourteen selected signaling partners, analyzed in TCGA uveal melanoma patients and uveal melanoma cell line datasets, revealed parallelism, with *GNAQ*, *ARHGEF11,* and *RHOJ* being among the highest expressed transcripts in both groups (Figure 6B). Among the components of the signaling landscape that accompanies the Gαq-Q209L/PDZ-RhoGEF/RhoJ signaling axis, those whose names are underlined in Figure 6B have been revealed as vulnerabilities in various cancer cell lines (Figure 6C), due to synthetic lethality strategies analyzed at the cancer dependency map datasets (https://depmap.org/portal/; accessed on 12 October 2023) [27]. To identify whether the transcriptional expressions of members of the signaling landscape that accompanies the Gαq-Q209L/PDZ-RhoGEF/RhoJ signaling axis were potentially linked with higher risk of shorter patient survival, we organized them, according to the signaling functions of the encoded proteins as GPCRs: *GPR21*, *GPR173,* and *EDNRA* (Figure 6D, upper panel); signaling effectors with catalytic properties, including kinases, phosphatases, and small GTPases (Figure 6D, second panel); and non-catalytic signaling proteins (Figure 6D, lower panel). We assessed, together with *GNAQ*, *ARHGEF11,* and *RHOJ*, their statistical correlations, as independent transcriptional signatures, with shorter patient survival. As shown in Figure 6E, the three transcriptional signatures exhibited significant statistical correlations with shorter patient survival. 

## 3. Discussion

We previously demonstrated that PDZ-RhoGEF, a well-known effector of Gα_13_ that activates RhoA, promoting stress fiber formation and agonist-dependent cell retraction [10,28,29], can be activated by constitutively active Gαs. This activation leads to filopodia-like sprouting by directly activating Cdc42 upon the interaction of active Gαs with the DH-PH catalytic module of PDZ-RhoGEF [13]. Our current results indicate that oncogenic Gαq-Q209L (a GTPase-deficient mutant) drives RhoJ activation via PDZ-RhoGEF. These findings are consistent with a mechanism by which Gαq-Q209L interacts with DH-PH domains of PDZ-RhoGEF, enabling this catalytic module to gain affinity for RhoJ. This interpretation is further supported by the effect of the PH domain, which, as an isolated construct, inhibited the interaction of Gαq-Q209L with the PDZ-RhoGEF DH-PH module, decreasing its binding to RhoJ-G33A. Based on our results, we propose a model showing that oncogenic Gαq-Q209L binds to the PDZ-RhoGEF’s DH-PH module, exerting an allosteric effect that enables this catalytic module to bind and activate RhoJ (Figure 7). This mechanism is consistent with the structural basis of p63RhoGEF activation by Gαq, which binds the catalytic tandem, causing an allosteric effect to activate the GEF [30].

As an effector of oncogenic Gαq-Q209L, PDZ-RhoGEF might constitute a protumoral signaling hub linked to the cytoskeletal effects driven by RhoJ, which is involved in tumor-induced angiogenesis, metastatic dissemination of cancer cells, and resistance to anti-cancer therapies in cells undergoing epithelial mesenchymal transition [14,15,16,18]. Therefore, signaling companions of the Gαq-Q209L/PDZ-RhoGEF/RhoJ signaling axis in uveal melanoma patients, as those shown in Figure 7, might represent potential targets and biomarkers with clinical relevance. To further explore this possibility, we identified those potential signaling partners in uveal melanoma TCGA patient datasets [26]. We looked for those that correlated with shorter patient survival and that were preferentially co-expressed in patients with high expressions of Gαq, PDZ-RhoGEF, and RhoJ, with the intent to identify potential clinically relevant transcriptional signatures. We identified a group of 14 potential signaling partners, including GPCRs, kinases, other catalytic signaling effectors, and non-catalytic signaling partners. We organized them into three groups, which, together with Gαq, PDZ-RhoGEF, and RhoJ, constitute transcriptional signatures that correlated with shorter patient survival. Some of these signaling elements have been mechanistically linked to cellular processes that characterize cancer progression. For instance, the endothelin type A receptor, encoded by *EDNRA*, drives ovarian cancer progression by promoting invadopodia formation through a β-arrestin/PDZ-RhoGEF-mediated mechanism [31], promotes colorectal cancer progression [32], and is associated with metastasis in patients with advanced bladder cancer [33]. Another example, *DYRK3*, a dual-specificity kinase, contributes to glioblastoma malignancy [34], and *RAP2A* increases the migration and invasion of osteosarcoma cell lines [35]. Although our experiments were focused on the effects of oncogenic Gαq, our findings raise the possibility that the Gαq/PDZ-RhoGEF/RhoJ signaling axis might be driven by G protein-coupled receptors. Our data mining strategy highlighted some interesting candidates that will be analyzed in future studies.

Preclinical investigations have characterized PDZ-RhoGEF, a multidomain signaling effector, as a critical participant in the progression of various cancer types, including glioblastoma and ovarian cancer [31,36]. The gene coding for this RhoGEF has been found to be amplified in human breast invasive carcinoma and lung adenocarcinoma [37]. Based on the current studies, and previous findings, as a signaling platform that integrates heterotrimeric G protein signaling, leading to adjustments in the actin cytoskeleton, PDZ-RhoGEF might serve as an effector to fine-tune dynamic adjustments of migrating cancer cells, depending on the signaling input and the small Rho GTPase being activated: RhoA, Cdc42, or RhoJ. These results contribute to explain the contrasting functional effects of different Rho GTPases, particularly Cdc42 and RhoJ, which, given their high homology, could be expected to be functionally redundant [38,39], and open new avenues to explore the regulation by oncogenic Gαq of RhoJ effectors, which has, so far, been linked to vascular endothelial growth factor receptor and Semaphorin E/Plexin D signaling [40]. In uveal melanoma cells, the oncogenic effect of Gαq-Q209L is, at least in part, mediated by Trio [38,39], and our results add PDZ-RhoGEF to the repertoire of oncogenic effectors, warranting further investigations pointing to their potential as drug targets of metastatic cancers [37]. Although our current results are consistent with the potential oncogenic role of the Gαq-Q209L/PDZ-RhoGEF/RhoJ axis, we should keep in mind that this emerging possibility warrants future investigations in preclinical models of uveal melanoma.

## 4. Materials and Methods

### 4.1. Cell Cultures and Plasmids

HEK293T and PAE cells were cultivated in DMEM medium (Dulbecco’s modified Eagle´s medium) with 10% FBS (Byproductos, Guadalajara, Jal. Mexico, 90020500) and antibiotics (Gibco, 15240-062) at 37 °C with 5% CO_2_ atmosphere. For the experiments, HEK293T cells were seeded on 1X poly-D-lysine-coated plates (6 wells or p60 dishes) and transfected 24 h later. The following plasmids were used in this work: pCEFL-GST, pCEFL-EGFP-PRG-DH-PH-CAAX, pCEFL-EGFP-p115-DH-PH-CAAX, pCEFL-EGFP-LARG-DH-PH-CAAX, pCEFL-AU1-PRG, pCEFL-HA-Gαi-Q205L, pCEFL-HA-Gαs-Q227L, pCEFL-HA-Gαq-Q209L, pCEFL-HA-Gα13-Q226L, pCEFL-HA-Gαq-WT, pCEFL-GST-PRG-DH-PH, pCEFL-GST-PRG-DH, pCEFL-GST-PRG-PH, pCEFL-GST-PRG-Linker, pCEFL-EGFP-PRG-Linker, pCEFL-GST-RhoJ-G33A, pCEFL-mCherry-RhoJ-T35N, and pCEFL-RhoJ-WT [13,21,41]. RhoJ-WT, RhoJ-Q79L, and RhoJ-T35N were kindly donated by Dr. Victoria Heath, University of Birmingham [17]. The plasmids pCEFL-EGFP-ARHGEF7-DH-PH-CAAX, pCEFL-EGFP-FARP2-DH-PH-CAAX, pCEFL-EGFP-FGD5-DH-PH-CAAX, pCEFL-EGFP-FGD6-DH-PH-CAAX, and pCEFL-EGFP-TUBA-DH-PH-CAAX were obtained through PCR amplification of the DH-PH catalytic modules defined with the SMART platform (http://smart.embl-heidelberg.de/, accessed on 12 October 2023). We used the following primers for each RhoGEF: ARHGEF7-BamHI-DH ataGGATCCGTGCTACAGAATATTTTAGAAACAG and ARHGEF7-EcoRI-PH ataGAATTCCGTGACCTTCGTTTGCTTCTG; FARP2-NheI-DH ataGCTAGCATAGTCAAAGAGATTCTCGCT and FARP2-EcoRI-PH ataGAATTCGGCTGCTTGGATCGCGGAGTTC; FGD5-BamHI-DH ataGGATCCATCGCACAGGAACTGCTATCT and FGD5-EcoRI-PH ataGAATTCGTCCTCAGGGAGGGCTCTGCTC; FGD6-BglII-DH ataAGATCTATTGCCAAGGAGATCATGAGC and FGD6-NheI-PH ataGCTAGCATACTCTTCTATTGCCCTGGA; TUBA-BamHI-DH ataGGATCCGTCATAGAAGAACTTCTTCAG and TUBA-NheI-BAR ataGCTAGCCACTTTGAGTAACGAAAGCAGC; and hFARP1-DH-5’BamHI ataGGATCCATAGCTAAGGAAGTGTCTACC and hFARP1-PH3’EcoRI ataGAATTCCGCCAGGTCAATGGCCATCTGG. Full-length cDNA coding for the following RhoGEFs were kindly donated by the indicated colleagues: FGD5 from Dr. Yoshiyuki Rikitake, Kobe University [42]; FGD6 (KIAA1316) and FARP2 (KIAA0793) from Dr. Silvio Gutkind, University of California San Diego; FARP1 from Dr. Harry Mellor, University of Bristol [43]; ITSN1 and ITSN2 from Dr. Susana de Luna, Centre de Regulació Genómica-CRG, Barcelona [44]; ARHGEF16 (also known as Ephexin4) from Dr. Hironori Katoh, Kyoto University [45]; AKAP13 (also known as AKAP-Lbc) from Dr. Dario Diviani, Université de Lausanne [46]; ARHGEF18 (also known as p114RhoGEF) from Dr. Takuji Tanoue, Kobe University [47]; ARHGEF7 (also known as β-Pix or COOL-1) from Dr. Richard Cerione, Cornell University [48]; and TUBA from Dr. Pietro De Camilli, Yale School of Medicine, New Haven [49]. Cells were transfected with Turbofect (Thermo Scientific, catalog R0531) according to the manufacturer’s instructions.

### 4.2. GST Pulldown

HEK293T cells, seeded in 6-well plates pretreated with 1X poly-D-lysine, were transfected with the plasmids indicated in the figure legends. Subsequently, 24 h later, they were serum-starved overnight before lysis. Cells were washed with 1x PBS and ice-cold lysis buffer was added to each dish on ice. Cell lysates were prepared, with 1.0 mL for 6–10 cm dishes and 0.5 mL for 6-well plates. Lysis buffer was composed of TBS-Triton (50 mM Tris, pH 7.5, 150 mM NaCl, 1% Triton X-100) with 5 mM EDTA; protease inhibitors: 1 mM phenylmethylsulfonyl fluoride (PMSF), 10 μg/mL leupeptin, and 10 μg/mL aprotinin; phosphatase inhibitors: 10 mM β-glycerophosphate, 1 mM NaF, and 1 mM sodium orthovanadate. Lysates were transferred to 1.5 mL tubes and centrifuged at 13,000 rpm for 10 min at 4 °C. Then, a fraction of total cell lysates (TCLs), 100 µL, was diluted with 4X Laemmli sample buffer containing β-mercaptoethanol, boiled for 5 min, centrifuged 5 min/13,000 rpm, and stored at −20 °C until used with pulldowns in Western blot analysis. The rest of the cell lysate was used for pulldown or immunoprecipitation assays. In experiments with Gq inhibitor (YM254890, 1 µM, Tocris catalog 7352), cells were treated for 2 h before lysis. Cell lysates for pulldown were incubated with 30 µL glutathione-sepharose (Bio-sciences AB, catalog 17-0756-05) at 4 °C on a rocking platform for 1 h. Afterwards, the beads were washed 3 times, with 1 mL lysis buffer each time, and centrifuged at 5000 rpm. Finally, beads were suspended in 1X Laemmli sample buffer containing β-mercaptoethanol, boiled for 5 min, centrifuged at 13,000 rpm for 10 min, and subjected to SDS-PAGE, followed by Western blotting.

### 4.3. Immunoprecipitation

Cell lysates were incubated with 1 µL of anti-PDZ-RhoGEF overnight at 4 °C. The next day, cell lysates were incubated with 30 µL G protein-sepharose beads (Millipore, catalog 16-266) on a rocking platform for 3 h to capture immunocomplexes. Then, beads were washed 3 times with 1 mL lysis buffer. Finally, beads were suspended in 40 µL sample buffer (containing β-mercaptoethanol), boiled for 5 min, centrifuged at 13,000 rpm for 10 min, and subjected to Western blot analysis.

### 4.4. Western Blot

Cell lysates, immunoprecipitates, and pulldowns were separated in SDS-PAGE gels and transferred to PVDF filters (Immobilon-P, Millipore, catalog no. IPV00010). Subsequently, filters were blocked with 5% milk in 1X TBS-Tween, washed 3 times with TBS-Tween, and incubated with the corresponding primary antibody solution in TBS-Tween. The primary antibodies used were obtained from the following sources: anti-Gαq (sc-392), EGFP (sc-9996), GST (sc-138), ERK2 (sc-154), all from Santa Cruz Biotechnology; PDZ-RhoGEF (Human Protein Atlas No. HPA014658, Sigma); RhoJ (Abcam, ab57584); pERK (phospho-ERK1/2 T202/Y204, Cell Signaling Technology, catalog no. 9191); and HA (Covance, catalog no. MMS-101P). Anti-mouse (KPL, catalog no. 074-1802) and anti-rabbit (KPL, catalog no. 074-1516) secondary antibodies were used. Western blots were revealed with a chemiluminescent substrate (Millipore, catalog WNKLS0500).

### 4.5. Morphological Analysis of Endothelial Cells 

Endothelial cells (PAE) were seeded on gelatin-coated coverslips. The next day, cells were transfected with TurboFect with the following plasmids: EGFP-CAAX, EGFP-PRG-DH-PH-CAAX, and/or mCherry-RhoJ-T35N, as indicated in the figure legends, depending on the experimental condition. After 48 h, cells were fixed with 4% PFA and photographed in a Leica confocal laser scanning microscope TCS SP8, using a 63X 1.4 oil immersion objective. Images were analyzed with FIJI-ImageJ software. Cells were considered positive for the presence of filopodia-like structures when they had at least nine of these finger-like protrusions.

### 4.6. Data Mining of Uveal Melanoma Patient and Cell Lines Datasets

Uveal melanoma TCGA patient transcriptomic datasets [26] were analyzed from the cBioPortal platform (https://www.cbioportal.org/; accessed on 18 October 2023) to identify the signaling repertoire encoded by transcripts co-expressed with *GNAQ* (coding for Gαq), *ARHGEF11* (coding for PDZ-RhoGEF), and *RHOJ*. We focused on the signaling repertoire that correlated with at least two of these genes, preferentially in patients expressing them at high levels. Patients were segregated depending on high and low *GNAQ*, *ARHGEF11,* and *RHOJ* expression. Co-expression lists were filtered for genes coding for proteins with conserved signaling domains (agonists, receptors, kinases, phosphatases, and other catalytic and non-catalytic signaling proteins). Within the group of patients with high *GNAQ*, *ARHGEF11,* and *RHOJ* expression, the transcripts with the highest Spearman’s correlations that had a difference of at least 0.05 compared with patients with low expression of *GNAQ*, *ARHGEF11,* and *RHOJ* were selected. mRNA expression (RSEM (batch normalized from Illumina HiSeq_RNASeqV2)) was assessed in uveal melanoma TCGA patients and in six uveal melanoma cell lines with mutant *GNAQ*. Vulnerabilities were searched in uveal melanoma, skin melanoma, and lung cancer cell lines using CRIPSR and RNAi gene effect. Essential genes were considered with a T-statistic value of -0.5 or less. Transcripts coding for GPCRs, catalytic signaling effectors, and non-catalytic signaling proteins, such as those containing protein–protein interaction domains, were individually subjected to Kaplan–Meier survival analysis. Those in which higher expression significantly correlated with shorter patient survival were assessed as transcriptional signatures, along with *GNAQ*, *ARHGEF11,* and *RHOJ*. Patients were split with *Auto-select best cutoff* and analyzed using multivariate Cox regression (https://kmplot.com/analysis/index.php?p=service&cancer=custom_plot#, accessed on 12 October 2023). The essentiality of the selected genes was investigated from the synthetic lethality datasets, analyzed from the cancer dependency map portal (https://depmap.org/portal/; accessed on 12 October 2023) [27].

### 4.7. Statistical Analysis

Quantitative data are presented as the means ± S.E. of at least three independent experiments, as indicated in the figure legends. Protein–protein interactions were confirmed with at least two independent and additional complementary experiments. Densitometric analysis of Western blots was performed with ImageJ software. Experimental results were normalized concerning the total amount of proteins. Statistical analysis and graphical representation of results were conducted with GraphPad Prism 10 software and are indicated in the figure legends.

## 5. Conclusions

In conclusion, we demonstrated that an oncogenic Gαq mutant (Gαq-Q209L) directly enables the PDZ-RhoGEF DH-PH catalytic module to activate RhoJ and identified elements of the related signaling landscape in TCGA uveal melanoma patients, which, as transcriptional signatures, correlated with shorter patient survival. Our findings suggest an allosteric mechanism by which oncogenic Gαq stimulates RhoJ via PDZ-RhoGEF and illuminate a new potential target axis in Gq-driven tumors.

## Figures and Tables

**Figure 1 ijms-24-15734-f001:**
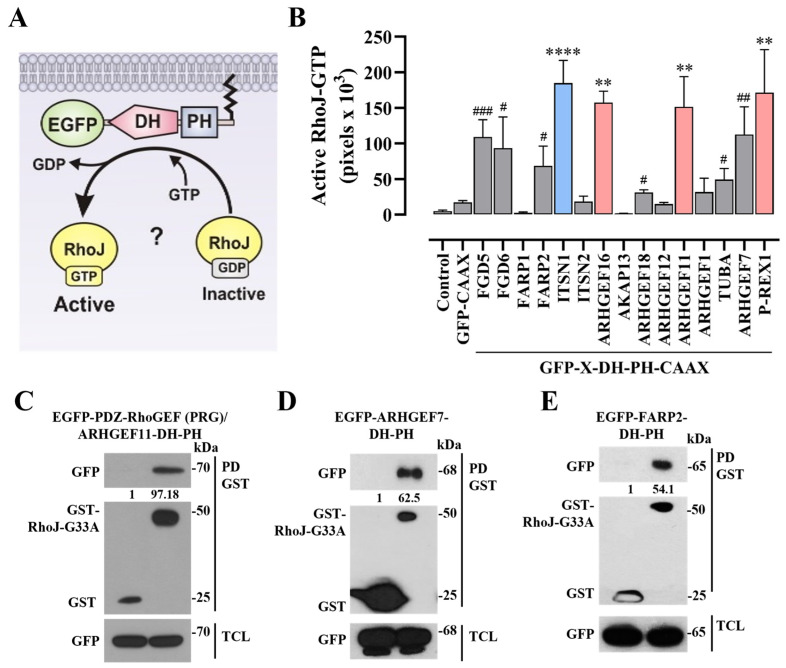
Screening of RhoJ activation induced by the catalytic tandem of different RhoGEFs. (**A**) Scheme of the EGFP-DH-PH-CAAX constructs of RhoGEFs; hypothetically, some of them activate RhoJ. (**B**) Screening of the DH-PH catalytic tandems of RhoGEFs in RhoJ activation. HEK-293T cells were transfected with RhoJ and EGFP-DH-PH-CAAX constructs from different RhoGEFs, and the cell lysates were incubated with PAK-CRIB beads to capture active RhoJ. The graph represents the mean activation of RhoJ by each constitutively active RhoGEF-DH-PH construct. Results are from 3 to 7 independent experiments. **** *p*<0.0001, ** *p*<0.01 vs. EGFP-CAAX. One-way ANOVA followed by Dunnet test; ITSN1 (blue bar) was used as the positive control [21]; red bars represent the RhoGEFs with the highest significant effects. ^###^
*p* < 0.001, ^##^
*p* < 0.01 ^#^
*p* < 0.05 vs. EGFP-CAAX; *t*-test. Interactions of EGFP-PDZ-RhoGEF-DH-PH (**C**), EGFP-ARHGEF7-DH-PH (**D**), and EGFP-FARP2-DH-PH (**E**), with co-transfected nucleotide-free RhoJ (GST-RhoJ-G33A). HEK-293T cells were transfected with the corresponding plasmids and subjected to GST pulldown assays. The numbers below the top panels (**C**–**E**) indicate the normalized densitometric value, with respect to GST, used as the control. Results are representative of at least 3 independent experiments.

**Figure 2 ijms-24-15734-f002:**
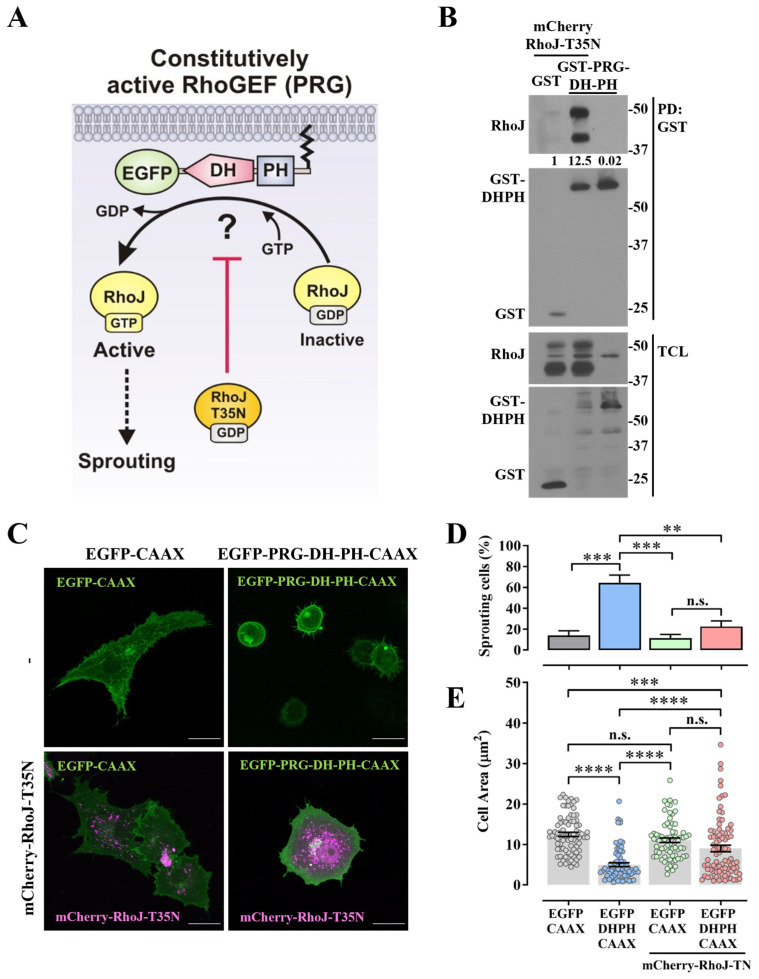
Dominant negative RhoJ interacts with PDZ-RhoGEF-DH-PH, preventing its morphological effects. (**A**) Hypothetical model of constitutively active construct of PDZ-RhoGEF (PRG) activates RhoJ and is sensitive to the inhibitory effect of a dominant negative RhoJ-T35N mutant. (**B**) Dominant negative RhoJ interacts with PRG DH-PH tandem. Transfected HEK-293T cells were subjected to a GST pulldown assay and mCherry-RhoJ-T35N was detected using Western blotting. The numbers below the top panel indicate the normalized densitometric value with respect to GST, used as the control. The result is representative of 3 independent experiments. (**C**) Effect of the dominant negative RhoJ-T35N mutant in the morphological effects caused by a constitutively active PRG construct (GFP-PRG-DH-PH-CAAX). Fluorescence of endothelial cells (PAE) transfected with EGFP-CAAX or EGFP-PRG-DH-PH-CAAX with or without mCherry-RhoJ-T35N. Images are representative of three independent experiments, in which at least 25 cells per experiment were analyzed. (**D**) Graph indicates the percentage of sprouting cells from three independent experiments; data are represented as the mean ± SEM, ** *p* < 0.01, *** *p* < 0.001, n.s., non significant. (**E**) Quantitative analysis of the morphological effects of GFP-PRG-DH-PH on the endothelial cell area in the presence or absence of the dominant negative RhoJ-T35N mutant, with EGFP-CAAX used as the negative control. Bars represent the mean ± SEM of at least 80 cells; *** *p* < 0.001, **** *p* < 0.0001, n.s. non significant; *n* = 3 experiments.

**Figure 3 ijms-24-15734-f003:**
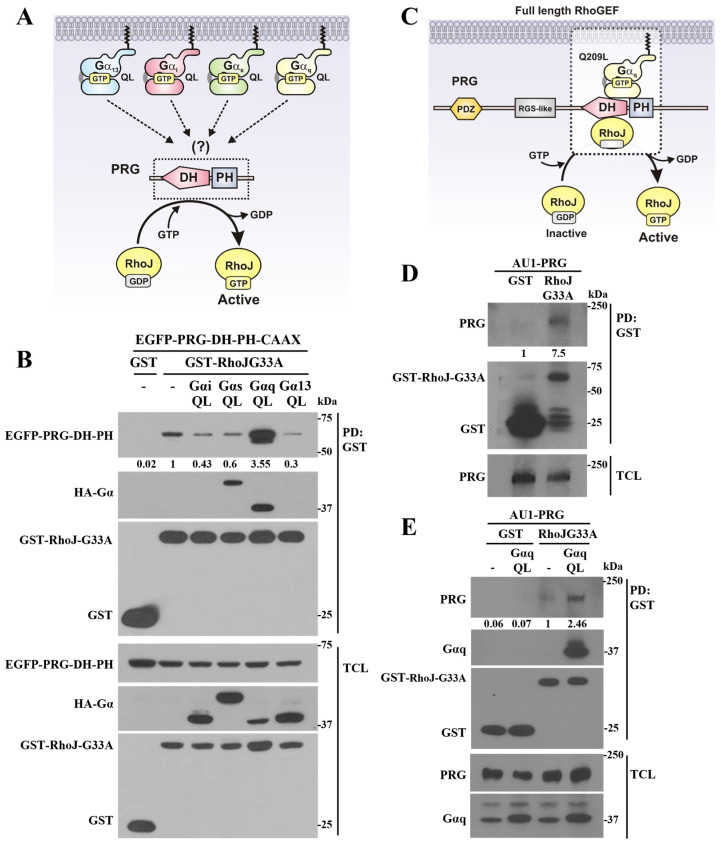
Constitutively active Gαq-QL enables PRG-DH-PH to gain affinity for RhoJ. (**A**) Hypothetical model showing the possibility that constitutively active Gα subunits of the four families of heterotrimeric G proteins (QL mutants) drive PRG-DH-PH to activate RhoJ. (**B**) Effect of different active Gα subunits (QL mutants) on the interaction between EGFP-PRG-DH-PH-CAAX and RhoJ-G33A. Lysates of transfected HEK-293T cells were subjected to GST pulldown assays, followed by the indicated Western blots. (**C**) Hypothetical model showing full-length PDZ-RhoGEF (PRG) as an effector of Gαq-Q209L to activate RhoJ-G33A. (**D**) Interaction of PRG with RhoJ-G33A, analyzed using a pulldown assay. (**E**) Effect of Gαq-Q209L on the interaction between PRG and RhoJ-G33A. Transfected HEK293T cells were subjected to pulldown assays, followed by Western blots, to detect full-length PRG, Gαq, and GST-RhoJ-G33A; GST was used as the negative control. Results shown in (**B**,**D**,**E**) are representative of at least 2 independent experiments. The numbers below the top panels (**B**,**D**,**E**) indicate the densitometric values with respect to the condition normalized as 1.

**Figure 4 ijms-24-15734-f004:**
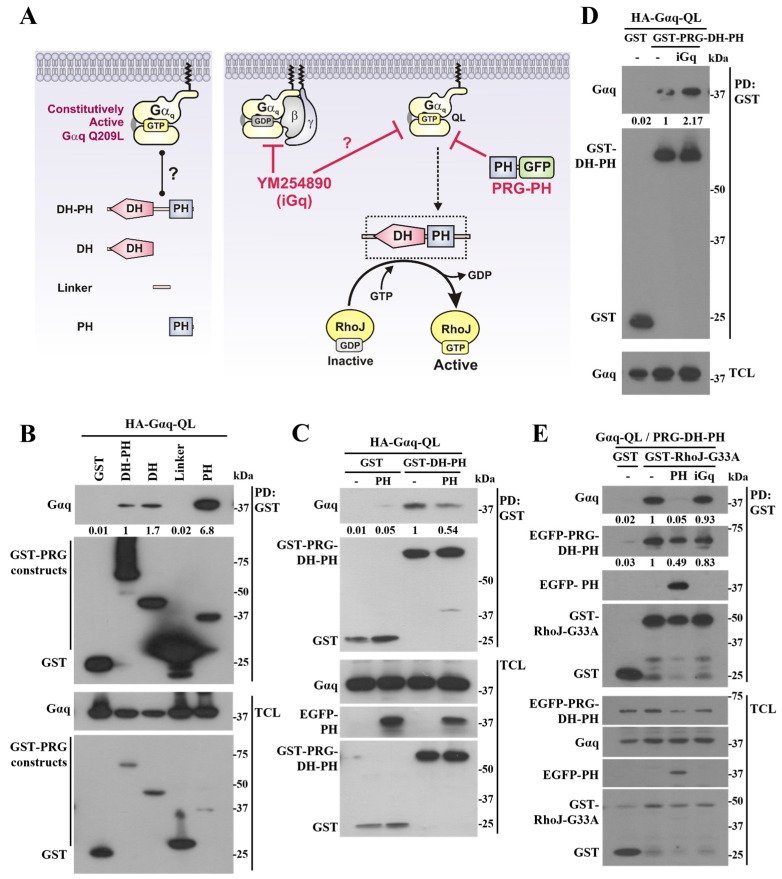
Constitutively active Gαq-Q209L mutant mainly interacts with the PRG-PH domain. (**A**) Left, hypothetical model showing the interaction of Gαq-Q209L with constructs obtained from the PRG-DH-PH catalytic tandem. Right, the potential effect of YM254890 (an inhibitor of GPCR-dependent activation of Gq and iGq) or the PRG-PH domain (as an EGFP-tagged construct) on the interaction between Gαq-Q209L and PRG-DH-PH. (**B**) Pulldown analysis of the interaction between Gαq-Q209L and the indicated PDZ-RhoGEF constructs. (**C**) Effect of the PRG-PH domain on the interaction between Gαq-Q209L and PRG-DH-PH. (**D**) Effect of YM254890 (iGq) on the interaction between Gαq-Q209L and PRG-DH-PH. (E) Effect of the EGFP-PRG-PH domain or YM254890 (iGq) on the ternary complex formed by Gαq-Q209L, PRG-DH-PH, and RhoJ-G33A. Experiments shown in B-E were conducted with transfected HEK-293T cells. Results are representative of at least 3 independent experiments. The numbers below the top panels (**B**–**E**) indicate the densitometric values with respect to the condition normalized as 1.

**Figure 5 ijms-24-15734-f005:**
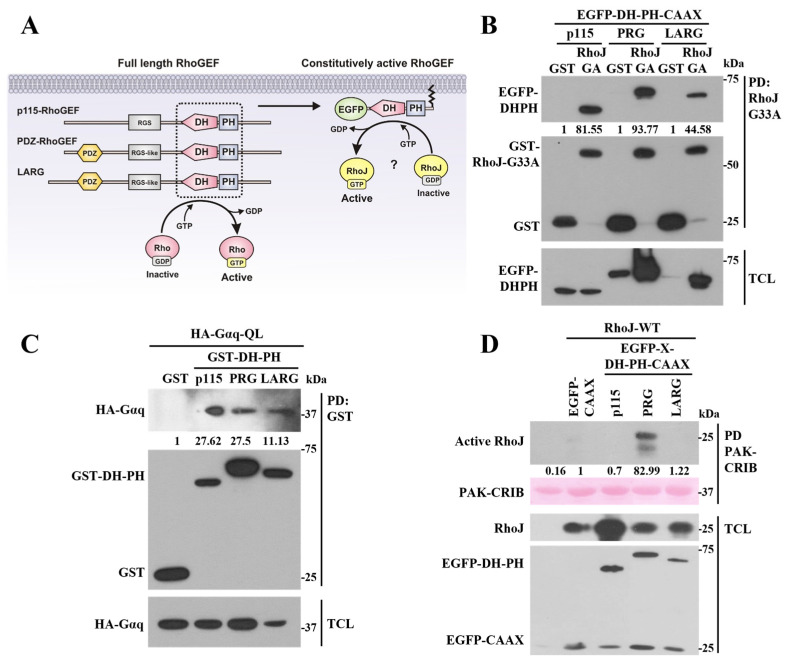
Gαq-QL interacts with the DH-PH module of all three RGS-RhoGEFs, but only PRG-DH-PH activates RhoJ. (**A**) Hypothetical model showing the constitutively active constructs of the RGS-RhoGEFs that potentially activate RhoJ. (**B**) Pulldown analysis assessing the interaction between constitutively active RGS-RhoGEF DH-PH constructs (p115, PRG, LARG) and GST-RhoJ-GA. (**C**) Analysis of the interaction between constitutively active Gαq-Q209L and p115, and PRG and LARG DH-PH catalytic tandems, fused to GST, addressed using pulldown, followed by Western blot, to detect co-transfected Gαq-Q209L; GST was used as the negative control. (**D**) Effect of constitutively active RGS-RhoGEF DH-PH constructs on the activation of RhoJ. HEK-293T cells transfected with wild-type RhoJ and the indicated RGS-RhoGEF DH-PH constructs were subjected to recombinant GST-PAK-CRIB pulldown assays, followed by Western blot, to detect the active fractions of RhoJ isolated in the pulldown assays. Expression of the transfected proteins was analyzed in total cell lysates (TCLs). The results shown in (**B**–**D**) are representative of at least 3 independent experiments, all of them conducted with transfected HEK-293T cells. The numbers below the top panels (**B**–**D**) indicate the densitometric values with respect to the condition normalized as 1.

**Figure 6 ijms-24-15734-f006:**
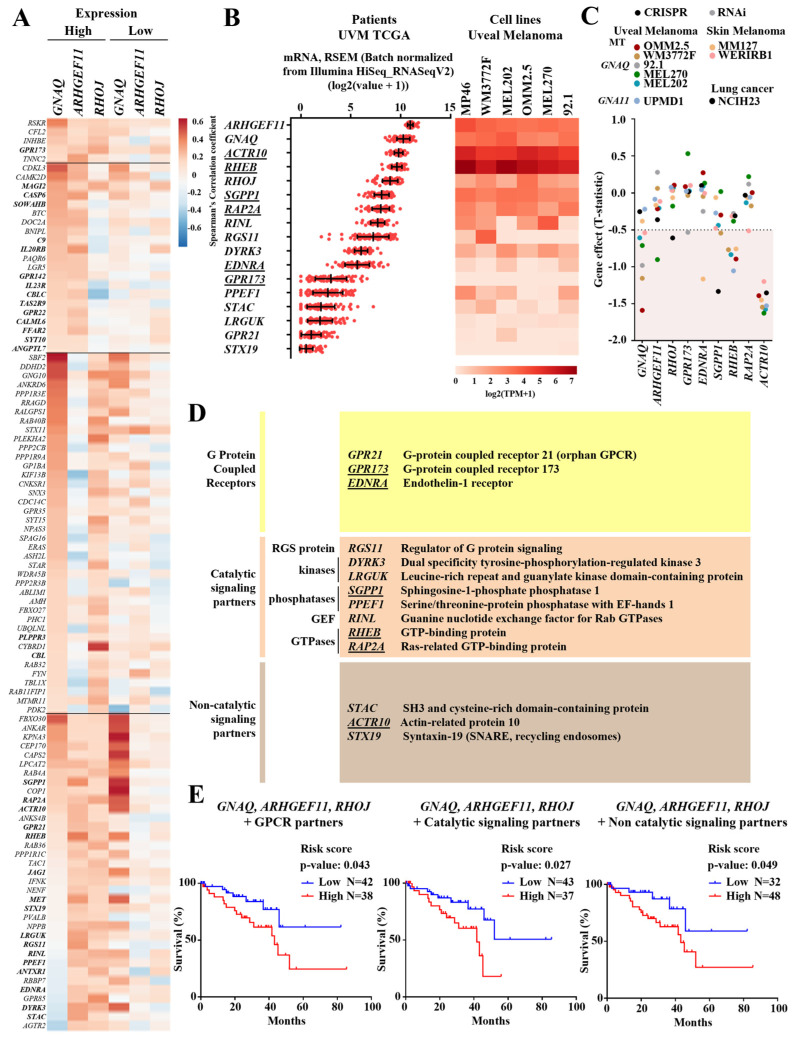
Data mining of TCGA uveal melanoma patient datasets and uveal melanoma cell lines reveals a group of Gαq/PRG/RhoJ signaling companions. (**A**) Co-expression analysis of the signaling landscape that accompanies the Gαq/PDZ-RhoGEF/RhoJ signaling axis. The heatmap represents the Spearman’s correlation coefficient of the 101 selected genes coding for signaling proteins, including receptors, kinases, and other catalytic effectors and non-catalytic signaling proteins that were correlated with GNAQ, ARHGEF11, and RHOJ in patients, segregated by low and high expression of these genes. Signaling partners were selected from the groups positively co-expressed with at least two of the three genes of the GNAQ, ARHGEF11, and RHOJ groups. Gene names in bold indicate those whose high expressions were correlated with shorter patient survival. (**B**) mRNA expressions of GNAQ, ARHGEF11, RHOJ and 14 signaling partners in uveal melanoma patients (left) and relevant cell lines (right). (**C**) CRISPR and RNAi effect of GNAQ, ARHGEF11, RHOJ and six coessential signaling companions. T-statistic cutoff value of −0.5. (**D**) Table of selected GPCRs, catalytic effectors, and non-catalytic signaling partners, mainly from the group that was correlated with ARHGEF11 and RHOJ, presented based on their function. Underlined names indicate genes that have been identified as essential in various types of cancer cell lines. (**E**) Signaling transcriptional signatures integrated by GNAQ, ARHGEF11, and RHOJ, and three groups of signaling partners were analyzed using multivariate Cox regression. The risk scores and survival curves are shown for the analysis of the signatures including GPCRs (**left**), as well as catalytic (**middle**) and non-catalytic (**right**) signaling partners. Patients were split with Auto-select best cutoff and analyzed using the KM plotter platform.

**Figure 7 ijms-24-15734-f007:**
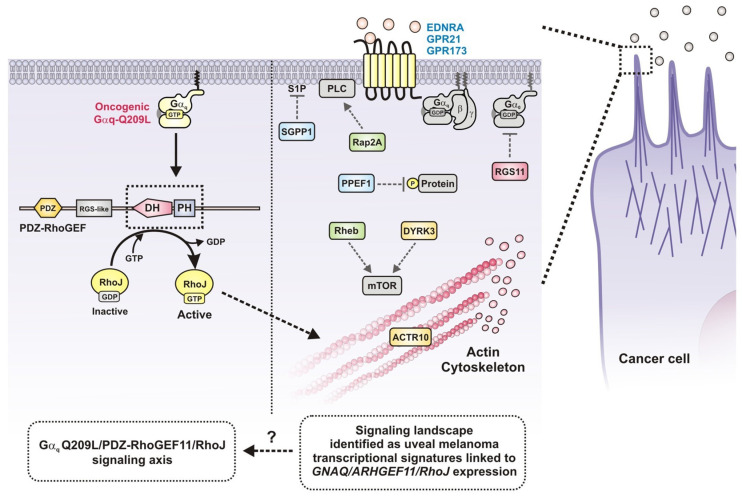
The Gq/PDZ-RhoGEF/RhoJ axis is accompanied by a signaling protein signature in uveal melanoma. Hypothetical model showing the cytoskeletal and morphological effects of oncogenic Gαq, which, via PDZ-RhoGEF, activates RhoJ. The signaling landscape that accompanies the Gq/PDZ-RhoGEF/RhoJ axis includes GPCRs, protein kinases and phosphatases, and small GTPases, raising a potential regulatory circuit that that warrants future characterization.

## Data Availability

All representative data are included in the manuscript. Original uncropped western blots and fluorescence images of cells are included in the supplemental materials.

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
