# Peer review of "Oncogenic Gαq activates RhoJ through PDZ-RhoGEF"

_ijms, 2023, doi:10.3390/ijms242115734_

Round 1

Reviewer 1 Report

Comments and Suggestions for Authors

The results presented in this manuscript by Cervantes-Villagrana et al indicate that oncogenic Gαq-Q209L (a GTPase-deficient mutant) drives RhoJ activation via PDZ-RhoGEF.

I found, that the topic is original and relevant in the field and addresses a specific gap in the field.

The methodology is fine and no further control is required.

I found the conclusion to be in line with the evidence and arguments presented.

The references are well-updated.

The manuscript is exceptionally well-written, and the presented results are highly compelling. Therefore, I  recommend accepting this manuscript for publication.

Reviewer 2 Report

Comments and Suggestions for Authors

Villagrana et al., in the study "Oncogenic Gq activates RhoJ via PDZ-RhoGEF," show that an oncogenic Gq mutant allows the PDZ-RhoGEF DH-PH module to recognize RhoJ, implying an allosteric mechanism by which this constitutively active GTPase stimulates RhoJ via PDZ-RhoGEF.  It is an interesting article with significant clinical implications in the field of uveal melanoma. There are a few questions that the author must address. The following are the queries:

1. The author employed HEK293 cells in all of his transfection studies. The mechanistic effect was demonstrated using HEK293 cells, but the same mechanism must be validated using uveal melanoma cell lines. Because of this, their study has less influence on readers.

2. Using TCGA data, the authors argue that genes connected to the G-alphaq/PDZ-Rho-GEF axis are associated with shorter survival. This will have a greater impact on readers if the authors evaluated the expression of those genes using UM cell lines like MP41, 92.1 and Mel202, which already carry the GNAQ-Q209L mutation.

3. The authors did not demonstrate how the introduction of these plasmids may affect the growth of cells. Does the introduction of these plasmids accelerate/decelerate oncogene-induced senescence since they are part of the oncogenic driver?

Reviewer 3 Report

Comments and Suggestions for Authors

Comments:

The manuscripts “Oncogenic Gαq activates RhoJ through PDZ-RhoGEF” by José Vázquez-Prado and et. al. identified three transcriptional signatures statisti-28 cally linked with shorter patient survival, including GPCRs and signaling effectors recognized as 29 vulnerabilities in cancer cell synthetic lethality datasets. In conclusion, we demonstrated that a 30 oncogenic Gαq mutant enables the PDZ-RhoGEF DH-PH module to recognize RhoJ, suggesting a 31 allosteric mechanism by which this constitutively active GTPase stimulates RhoJ via PDZ-RhoGEF. 32  

In my opinion, the paper is interesting.  I believe this work will have a high impact and will be of interest for IJMS readers. Consequently, I recommend the acceptance of this work in your respected journal after minor revisions

I have the following comments on the manuscript

-         Revise the whole manuscript for English and typographical errors

-        The rational of work should be more deeply explained

-         Add conclusion of work

-        Use more recent references

Comments on the Quality of English Language

minor revision

Reviewer 4 Report

Comments and Suggestions for Authors

The manuscript by Servantes-Villagrana et al is focused on the signaling pathway mediated by a trimeric G-protein, a GEF and a small GTPase. The authors found that G-alpha-q facilitates the interaction of PDZ-RhoGEF with RhoJ. This pathways appears to induce contraction of endothelial cells and formation of cell sprouts. Transcripts that simultaneously correlated with the expression of the three members of G-alpha-q/PDZ-RhoGEF/RhoJ axis and shorter survival of the patients with uveal melanoma have been identified through data mining.

The study is focused on an interesting and novel finding, the data are mostly convincing, and the results are of potential clinical significance. There are sev real issues, though, that can be addressed to improve the manuscript.

1. Fig. 1 and throughout the results - quantification of pull-down/immunoblotting data is needed. In many cases the result is not in the 'all-or-nothing' range, but requires a clear answer as to what is the difference between GST alone vs GST-fused protein (Fig. 1C, 2B, 3D, etc.), different proteins (Fig. 3B, 4B, etc), control and inhibitor. It is especially true when the expression levels for different interacting proteins vary.

2. The term 'signaling signature' is very vague. It is better to express this finding more precisely. Also, it would be useful to see a table with the selected partner proteins to have an easier look at their identity and functions, not just their gene name and grouping as GPCR, catalytic or non-catalytic. If the authors have a concrete idea as to how these partners  are involved in the G-alpha-q/PDZ-RhoGEF/RhoJ axis, a scheme should be presented graphically.

3. What were the lysis conditions (buffer, shaking/rotation, time, temperature)?

4. In Fig. 1B bars are color-coded. This is not explained in the legend.

5. Fig. 4D does not seem to be referenced in the text.

Round 2

Reviewer 2 Report

Comments and Suggestions for Authors

The authors have addressed most concerns raised in the previous review. I believe this has substantially improved the manuscript.